# Emotional Status and Psychological Well-Being in the Educational Opposition Process

Eduardo Melguizo-Ibáñez [1], Javier Cachón-Zagalaz [2], Gabriel González-Valero [1], Pilar Puertas-Molero [1,*], Laura García-Pérez [1] and José Luis Ubago-Jiménez [1]

1   Department of Didactics of Musical, Plastic and Corporal Expression, Faculty of Education Sciences, University of Granada, 18071 Granada, Spain; emelguizo@ugr.es (E.M.-I.); ggvalero@ugr.es (G.G.-V.); lgperez@ugr.es (L.G.-P.); jlubago@ugr.es (J.L.U.-J.)
2   Department of Didactics of Musical, Plastic and Corporal Expression, Faculty of Humanities and Educational Sciences, University of Jaén, 23071 Jaén, Spain; jcachon@ujaen.es
*   Correspondence: pilarpuertas@correo.ugr.es

**Abstract:** The process to become a public teacher in Spain involves a very strict examination. In order to pass this exam, a high level of emotional competence is required. During the preparation for this test, symptoms related to anxiety, depression and stress are experienced. The aim of this study was to investigate the effect of negative emotional state on emotional intelligence and psychological wellbeing as a function of the number of sittings in the competitive examination process. The research design was quantitative and exploratory. The sample consisted of 3578 candidates. The results show an increase in the effect of negative emotional states on emotional intelligence and psychological well-being in candidates who have taken part in the selection process more than twice. It is concluded that negative emotional states increase in their effect on emotional and psychological well-being as the number of exams taken increases.

**Keywords:** emotional status; psychological well-being; pre-service teachers; educational process





## 1. Introduction

In different European societies, the role played by teachers is fundamental to understanding the educational situation in each country (Parker et al. 2022). The established process for becoming a schoolteacher in Spain is different from that in the rest of Europe (Melguizo-Ibáñez et al. 2023). It consists of passing a test divided into two parts (Melguizo-Ibáñez et al. 2023). The first test consists of the demonstration of theoretical knowledge (RD 270/2022 2022). It is assessed by means of a written test in which the candidate must demonstrate theoretical knowledge and its application to a practical situation (RD 270/2022 2022). The second part of the test is based on the verification of the candidates' pedagogical aptitude (RD 270/2022 2022). This involves the presentation and defense of a learning case in the specialty for which the applicant is applying (RD 270/2022 2022).

### 1.1. Negative Emotional States during the Evaluation Process

The preparation for this test causes applicants to start this process with high expectations and a high level of self-demand (Calderón et al. 2020). Teachers' self-imposed self-demand leads to the emergence of negative emotional states (Zurita-Ortega et al. 2018). This leads to a decrease in motivation towards academic tasks (Cachón-Zagalaz et al. 2022), directly affecting test performance (De la Fuente and Amate 2019). Low test performance has been shown to lead to mental fatigue, stress and anxiety (Agyapong et al. 2022). Moreover, the more an assessment is repeated, the higher the levels of anxiety and stress reach (Amate-Romera and De la Fuente 2021; Gallego et al. 2016).

## 1.2. Anxiety and Stress in the Teaching Evaluation Process

Stress and anxiety are common states present during preparation for any assessment (Hernández-Ballester et al. 2023). The occurrence of academic stress is characterised by trying to achieve academic performance above the level of proficiency (Melguizo-Ibáñez et al. 2023). This fosters the onset of physical disorders and mental fatigue (Ozamiz-Etxebarría et al. 2021). If the state of over-performance lasts over time, anxiety levels may increase (Zhen et al. 2022). Anxiety causes muscular tension, restlessness and insecurity about a task (Zhen et al. 2022). Students with high levels of anxiety and stress perform less well when preparing for a test (Yusefzadeh et al. 2019). This can lead to a state of depression resulting from poor performance (De la Fuente and Amate 2019). If the state of depression lasts over time, it can act on the mental health of the applicants (Granero-Gallegos et al. 2023).

## 1.3. Emotional Intelligence in the Teaching Profession

The study of emotional intelligence has become increasingly relevant in the area of education (Gómez-Leal et al. 2022). Being emotionally competent has numerous benefits for mental health (Turner and Stough 2020). Emotional intelligence is the ability to perceive, value, express and understand one's own and others' emotions (Goleman 1995). It is also known as the ability to regulate and generate emotional states that favor emotional and intellectual thinking (Bru-Luna et al. 2021). There are a large number of paradigms for the study of emotional intelligence. The model proposed by Goleman (1995) results from the combination of emotional and personal aspects. Through this model, five fundamental skills are highlighted: awareness and identification of emotional states, emotional control, self-motivation, recognition of others' emotional states, and control of social relations and external emotions (Bru-Luna et al. 2021). This influences the emergence of emotional self-awareness, self-management of emotional states, social awareness and management of peer relationships (Goleman 1995). This model demonstrates that emotional intelligence is directly implicated in teacher well-being (Ellis et al. 2020).

## 1.4. Psychological Well-Being

Well-being has been studied from hedonistic and eudemonistic perspectives. Under the first perspective, well-being is studied under a vision based on people's environment and satisfaction level (Tomás et al. 2016). The eudemonistic view does not relate to the experience of positive situations or states (Tomás et al. 2016). It is related to the satisfaction of basic psychological needs and the achievement of goals established by the subjects (Corcoran and Flaherty 2022). Ryff and Keyes (1995) developed a model of psychological well-being consisting of six factors: self-acceptance, positive relationships, autonomy, mastery of the environment, purpose in life and personal growth. It is focused on the eudemonism perspective (Ryan and Deci 2001). In the field of education, teachers with greater emotional competence show a higher level of psychological well-being (Puertas-Molero et al. 2019). Conversely, anxiety, depression and stress have a negative impact on psychological well-being (Luque-Reca et al. 2022). In some cases, if these three states persist over time, a decrease in well-being is observed (Luque-Reca et al. 2022).

Our research objectives are as follows:

**O.1.** *To study the levels of psychological well-being, emotional intelligence and disruptive states (anxiety, depression and stress) in a sample of candidates for competitive examinations in infant and primary education.*

**O.2.** *To investigate the effect of disruptive emotional states (anxiety, depression and stress) on emotional intelligence and psychological wellbeing as a function of the number of competitive examinations taken.*

The research hypotheses are set out below:

**H1.** *Participants who have taken the competitive examination more than twice will show a greater effect of disruptive states on psychological well-being and emotional intelligence.*

**H2.** *Candidates who have sat fewer than twice in the teachers' examination process will show a lower effect of disruptive states on psychological well-being and emotional intelligence.*

## 2. Materials and Methods

### 2.1. Design

This research was performed with an exploratory, cross-sectional and ex post facto (non-experimental) design. No manipulation of any variable was involved. Data were collected only from a single group.

### 2.2. Participants

Convenience sampling was used to gather the sample data. The final sample consisted of 3578 candidates for the public teaching corps. The sample was heterogeneous. The sample consisted of 1798 women (50.25%) and 1780 men (49.75%). The participants' ages were from 24 to 61 years (M = 32.78; SD = 11.78). A total of 2147 participants had taken the test between 0 and 2 times (60%), while 1432 subjects had taken the test more than 3 times (40%). Table 1 shows the distribution of the population according to autonomous community.

**Table 1.** Sample distribution by Spanish region.

|  | N | % |
|---|---|---|
| Balearic Island | 8 | 0.2% |
| Melilla | 8 | 0.2% |
| Basque Country | 16 | 0.4% |
| La Rioja | 20 | 0.6% |
| Navarra | 34 | 1.0% |
| Cataluña | 38 | 1.1% |
| Cantabria | 78 | 2.2% |
| Asturias | 129 | 3.6% |
| Canary Island | 131 | 3.7% |
| Murcia | 139 | 3.9% |
| Extremadura | 147 | 4.1% |
| Castilla La Mancha | 225 | 6.3% |
| Galicia | 233 | 6.5% |
| Aragón | 280 | 7.8% |
| Castilla y León | 336 | 9.4% |
| Valencian Community | 449 | 12.5% |
| Madrid | 554 | 15.5% |
| Andalucía | 745 | 20.8% |

A single inclusion criterion was established: being an applicant to the Spanish education system.

A total of 4327 responses were collected, but 730 were eliminated. These were deleted because the subjects did not meet the inclusion criterion. To guarantee that the questions were not randomly answered, two questions were duplicated. When the results of these did not match, participants were eliminated. In summary, 19 responses were rejected. For sampling error, a confidence level of 95% was established, resulting in a margin of error of less than 5.0%.

### 2.3. Participants

**Own ad hoc questionnaire:** This was used to collect the participants' age, gender (male/female), autonomous community and number of calls for applications to the competitive examination process (Melguizo-Ibáñez et al. 2023).

**Depression, Anxiety and Stress Scale** (Lovibond and Lovibond 1995): The Spanish version (Daza et al. 2002) was applied for this study. It is made up of 21 items (e.g., "I found it difficult to work up the initiative to do things") that allow the evaluation of the three negative emotional states. The depression subscale is composed of items 3, 5, 10, 13, 16, 17 and 21. The anxiety subscale is composed of items 2, 4, 7, 9, 15, 19 and 20. The stress subscale is composed of items 1, 6, 8, 11, 12, 14 and 18. Cronbach's alpha test for the depression, anxiety and stress subscales obtained values of $\alpha = 0.881$, $\alpha = 0.816$ and $\alpha = 0.840$, respectively.

**Psychological Well-Being Scale** (Ryff 1989): The version by Díaz et al. (2006) was used. It consists of a total of 39 items (e.g., "People would describe me as a giving person, willing to share my time with others"). This questionnaire assesses psychological well-being through six sub-variables: self-acceptance (items 1, 7, 17 and 24) ($\alpha = 0.850$), positive relationships (items 2, 8, 12, 22 and 25) ($\alpha = 0.774$), autonomy (items 3, 4, 9, 13, 18 and 23) ($\alpha = 0.730$), mastery of the environment (items 5, 10, 14, 19 and 29) ($\alpha = 0.811$), personal growth (items 21, 26, 27 and 28) ($\alpha = 0.709$) and purpose in life (items 6, 11, 15, 16 and 20) ($\alpha = 0.702$).

**Trait Emotional Intelligence Questionnaire—Short Form** (Cooper and Petrides 2010): This is an instrument consisting of 30 items (e.g., "On the whole, I am able to deal with stress") that are assessed through a Likert scale. Emotional intelligence is assessed through six dimensions: emotionality (items 1, 2, 8, 13, 16, 17, 23, 28), self-control (items 4, 7, 15, 19, 22, 30), well-being (items 5, 9, 12, 20, 24, 27), sociability (items 6, 10, 11, 21, 25, 26), self-motivation (items 3 and 18) and adaptability (items 14 and 29). Cronbach's alpha test evidenced the following scores for each of the subscales: emotionality ($\alpha = 0.833$), self-control ($\alpha = 0.850$), well-being ($\alpha = 0.750$), sociability ($\alpha = 0.901$), self-motivation ($\alpha = 0.865$) and adaptability ($\alpha = 0.849$).

### 2.4. Procedure

A review of the scientific literature was carried out to identify the most reliable instruments. Subsequently, a Google Form was elaborated with the instruments described above. Once the questionnaire was created, the research was promoted on the social networks of the University of Granada. Through this medium, it was possible to reach a large number of participants. Different preparers of educational competitive examinations were contacted and offered to share the questionnaire. The questionnaire was available from January to June 2022. The entrance exam was held in June 2022.

Regarding the ethical principles governing this research, the research complied with the Declaration of Helsinki. This investigation was also approved and monitored by the ethical committee of the University of Granada 2966/CEIH/2022.

### 2.5. Data Analysis

The IBM SPSS Statistics 25.0 statistical program was used to analyse the data. The reliability of the instruments was studied using Cronbach's alpha test. The distribution of the sample was analysed using the Kolmogorov–Smirnov test. This analysis indicated a normal distribution of the sample. Parametric tests were used. Pearson's bivariate correlations test was used to carry out the correlational analysis.

IBM SPSS Amos 23.0 software was used to develop the structural equation models. The recommendations established in the scientific literature (Maydeu-Olivares 2017; Kyriazos 2018) were followed for model fitting and evaluation. Of the absolute fit indices, the Chi-Square/degrees of freedom ($\chi^2$/gL) is the most common (Hu and Bentler 1999). It is the most sensitive to sample size (Hu and Bentler 1999). When a multi-group model is

presented, it is necessary that the sample size in each group is not too small (Hu and Bentler 1999). If this index indicates a value of less than three, a good fit is shown (Ruiz et al. 2010).

For comparative fit indices, the most commonly used are the goodness-of-fit index (CFI), Tucker–Lewis index (TLI), normalised fit index (NFI), goodness-of-fit index (GFI) and adjusted goodness-of-fit index (AGFI). The values of these indices must be greater than 0.900 (Hu and Bentler 1999). It is advisable to use other fit indices such as the root-mean-squared residuals of approximation (RMSEA). This value should be less than 0.100 (Maydeu-Olivares 2017). Each model was made up of a total of 18 variables. It was observed that 17 variables were endogenous and 1 was exogenous. In figures and tables, the arrows symbolise the directions in which the effects occur. The significance level was set at $p \leq 0.05$.

Two theoretical models have been put forward. The first one (Figure 1) presents a total mediation of emotional intelligence. The effect of negative emotional states is indirect on psychological well-being. The second model (Figure 1) presents a partial effects model. It assumes a direct effect of negative emotional states on psychological well-being. The fit values of the described indices show a better fit for the partial medication model (Table 2).

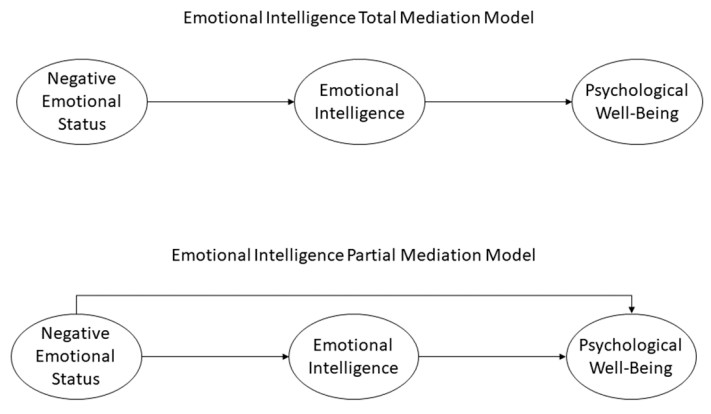

**Figure 1.** Theoretical models tested.

**Table 2.** Fit indices of the proposed models.

| | $\chi^2/gL$ | RMSEA | CFI | AGFI | GFI | CFI | TLI | NFI |
|---|---|---|---|---|---|---|---|---|
| **Model 1** | 3.846 ($p$ = 0.089) | 0.081 | 0.889 | 0.889 | 0.903 | 0.900 | 0.887 | 0.837 |
| **Model 2** | 2.189 ($p$ = 0.037) | 0.060 | 0.905 | 0.946 | 0.927 | 0.915 | 0.902 | 0.885 |

## 3. Results

For the achievement of Objective 1, Table 3 shows the mean values, standard deviations and correlational analysis results for the variables under study. Anxiety, depression and stress correlate positively with each other. All the variables that make up emotional intelligence correlate positively with each other. A correlational analysis of psychological well-being showed positive correlations between almost all variables. Negative relationships were observed with self-acceptance (r = −0.169; $p < 0.01$), self-control (r = −0.083; $p < 0.01$) and personal growth (r = −0.179; $p < 0.01$). Depression correlates negatively and with statistically significant differences with most variables. Negative correlations were observed with emotionality (r = 0.372; $p < 0.01$), self-motivation (r = 0.164; $p < 0.01$), adaptability (r = 0.184; $p < 0.01$) and personal relationships (r = 0.311; $p < 0.01$). Next, anxiety correlates negatively and significantly with most of the variables that make up emotional intelligence and psychological well-being. Positive correlations of anxiety were observed with emotionality (r = 0.267; $p < 0.01$). Finally, stress correlates positively only with emotionality (r = 0.267; $p < 0.01$).

**Table 3.** Mean and correlational analysis of the variables.

| | M | SD | Negative Emotional Status | | | | Emotional Intelligence | | | | | | Psychological Well-Being | | | | |
|---|---|---|---|---|---|---|---|---|---|---|---|---|---|---|---|---|---|
| | | | 1 | 2 | 3 | 4 | 5 | 6 | 7 | 8 | 9 | 10 | 11 | 12 | 13 | 14 | 15 |
| DP (1) | 1.034 | 0.716 | - | 0.587 ** | 0.578 ** | 0.372 ** | −0.128 ** | −0.227 ** | −0.036 * | 0.164 ** | 0.184 ** | −0.297 ** | 0.311 ** | −0.325 ** | −0.048 ** | −0.606 ** | −0.537 ** |
| AN (2) | 1.293 | 0.665 | | - | 0.707 ** | 0.267 ** | −0.159 ** | −0.123 ** | −0.002 ** | −0.057 ** | −0.124 ** | −0.123 ** | −0.179 ** | −0.079 ** | −0.031 ** | −0.324 ** | −0.276 ** |
| ST (3) | 1.828 | 0.618 | | | - | 0.310 ** | −0.202 ** | −0.114 ** | −0.078 ** | −0.090 ** | −0.177 ** | −0.149 ** | −0.203 ** | −0.065 ** | −0.053 ** | −0.303 ** | −0.299 ** |
| EM (4) | 3.591 | 0.615 | | | | - | 0.134 ** | 0.07 1** | 0.091 ** | 0.226 ** | 0.263 ** | −0.046 ** | 0.256 ** | −0.107 ** | 0.211 ** | −0.225 ** | −0.173 ** |
| SC (5) | 3.877 | 0.670 | | | | | - | 0.260 ** | 0.314 ** | 0.060 ** | 0.082 ** | 0.204 ** | 0.037 ** | 0.156 ** | 0.158 ** | 0.131 ** | 0.187 ** |
| WB (6) | 4.173 | 0.607 | | | | | | - | 0.258 ** | 0.126 ** | 0.109 ** | 0.365 ** | −0.002 ** | 0.361 ** | 0.257 ** | 0.408 ** | 0.429 ** |
| SO (7) | 4.074 | 0.613 | | | | | | | - | 0.118 ** | 0.127 ** | 0.172 ** | 0.029 ** | 0.157 ** | 0.138 ** | 0.110 ** | 0.207 ** |
| SM (8) | 4.012 | 0.677 | | | | | | | | - | 0.187 ** | 0.008 ** | 0.100 ** | −0.016 ** | 0.100 ** | −0.085 ** | −0.068 ** |
| AD (9) | 4.287 | 0.899 | | | | | | | | | - | 0.055 ** | 0.104 ** | 0.002 ** | 0.101 ** | −0.075 ** | −0.006 ** |
| SA (10) | 3.895 | 0.585 | | | | | | | | | | - | −0.169 ** | 0.420 ** | 0.197 ** | 0.463 ** | 0.492 ** |
| PR (11) | 3.543 | 0.713 | | | | | | | | | | | - | −0.083 ** | 0.230 ** | −0.179 ** | −0.200 ** |
| AU (12) | 4.186 | 0.606 | | | | | | | | | | | | - | 0.305 ** | 0.526 ** | 0.462 ** |
| ME (13) | 3.873 | 0.535 | | | | | | | | | | | | | - | 0.249 ** | 0.267 ** |
| PG (14) | 4.709 | 0.220 | | | | | | | | | | | | | | - | 0.650 ** |
| PL (15) | 4.328 | 0.657 | | | | | | | | | | | | | | | - |

**Note:** Depression (DP); anxiety (AN); stress (ST); emotionality (EM); self-control (SC); well-being (WB); sociability (SO); self-motivation (SM); adaptability (AD); self-acceptance (SA); positive relationships (PR); autonomy (AU); mastery of environment (ME); personal growth (PG); purpose in life (PL). **Note:** * Correlation is significant at the the 0.05 level (bilateral), ** Correlation is significant at the 0.01 level (bilateral).

Figure 2 and Table 4 show the theoretical partial mediation model together with the standardised regression weights (Objective 2 and Hypotheses 1 and 2). The first model refers to the participants who have submitted between zero and two times to the opposition process.

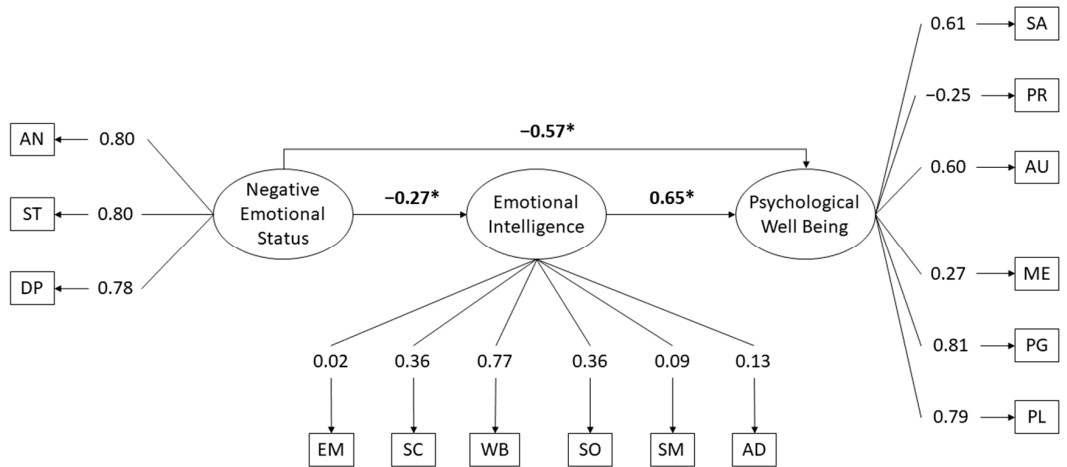

**Figure 2.** Theoretical model for participants who have presented 0 to 2 times. **Note:** Depression (DP); anxiety (AN); stress (ST); emotionality (EM); self-control (SC); well-being (WB); sociability (SO); self-motivation (SM); adaptability (AD); self-acceptance (SA); positive relationships (PR); autonomy (AU); mastery of environment (ME); personal growth (PG); purpose in life (PL). * $p < 0.01$.

**Table 4.** Standardised regression weights for participants who have applied between 0 and 2 times.

| Associations between Variables | RW | | | | SRW |
|:---:|:---:|:---:|:---:|:---:|:---:|
| | Estimations | SE | CR | *p* | Estimations |
| AN ← NES | 1.000 | | | | 0.803 |
| ST ← NES | 0.921 | 0.020 | 46.258 | ≤0.001 | 0.799 |
| DP ← NES | 1.000 | 0.023 | 44.334 | ≤0.001 | 0.782 |
| AD ← EI | 1.000 | | | | 0.131 |
| SM ← EI | 0.705 | 0.141 | 5.001 | ≤0.001 | 0.089 |
| SO ← EI | 1.788 | 0.260 | 6.881 | ≤0.001 | 0.363 |
| WB ← EI | 3.722 | 0.524 | 7.097 | ≤0.001 | 0.771 |
| SC ← EI | 1.916 | 0.279 | 6.863 | ≤0.001 | 0.360 |
| EM ← EI | 0.332 | 0.104 | 3.181 | 0.049 | 0.022 |
| SA ← PSWB | 1.000 | | | | 0.612 |
| PR ← PSWB | −0.415 | 0.037 | −11.190 | ≤0.001 | −0.250 |
| AU ← PSWB | 1.055 | 0.036 | 29.569 | ≤0.001 | 0.601 |
| ME ← PSWB | 0.489 | 0.029 | 17.141 | ≤0.001 | 0.270 |
| PG ← PSWB | 2.795 | 0.079 | 35.375 | ≤0.001 | 0.811 |
| PL ← PSWB | 1.492 | 0.042 | 35.228 | ≤0.001 | 0.791 |
| EI ← NES | −0.015 | 0.003 | −5.700 | ≤0.001 | −0.270 |
| PSWB ← EI | 0.029 | 0.004 | 6.878 | ≤0.001 | 0.653 |
| PSWB ← NES | −0.099 | 0.005 | −20.196 | ≤0.001 | −0.570 |

**Note:** Depression (DP); anxiety (AN); stress (ST); emotionality (EM); self-control (SC); well-being (WB); sociability (SO); self-motivation (SM); adaptability (AD); self-acceptance (SA); positive relationships (PR); autonomy (AU); mastery of environment (ME); personal growth (PG); purpose in life (PL). **Note:** Regression weights (RW); standardised regression weights (SRW); standard error (SE); critical ratio (CR).

Negative emotional states were shown to have a detrimental effect on emotional intelligence ($\beta = -0.27$; $p < 0.01$). A negative effect of these emotional states on emotional psychological well-being was shown ($\beta = -0.57$; $p < 0.01$). A positive effect of emotional intelligence on psychological well-being is evident ($\beta = 0.65$; $p < 0.01$).

Figure 3 and Table 5 present the theoretical model with standardised regression weights for participants who have gone through the process more than three times. A detrimental effect of negative emotional states on emotional intelligence was found ($\beta = -0.03$; $p < 0.01$). A detrimental effect of negative emotions on psychological well-being was shown ($\beta = -0.57$; $p < 0.01$). A positive effect of emotional intelligence on psychological well-being is evident ($\beta = 0.49$; $p < 0.01$).

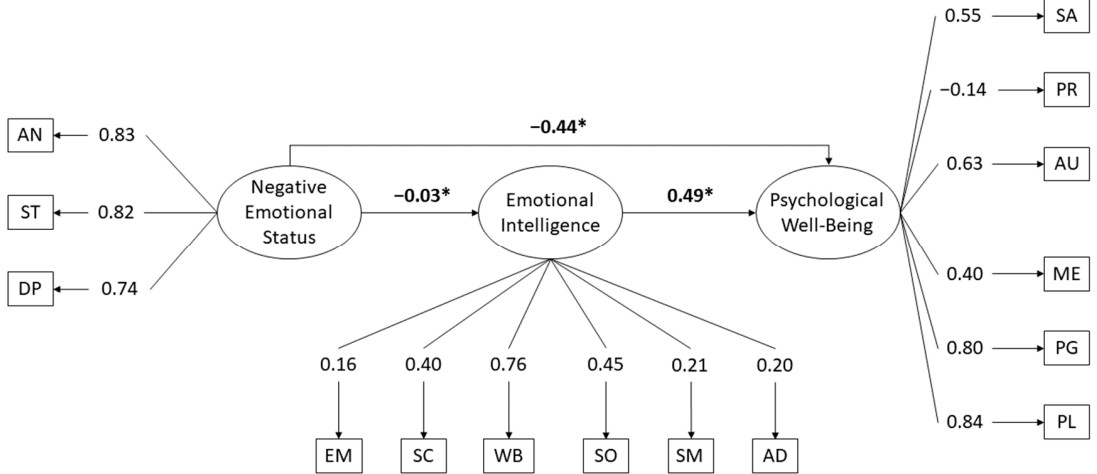

**Figure 3.** Theoretical model for participants who have presented 3 or more times. **Note:** Depression (DP); anxiety (AN); stress (ST); emotionality (EM); self-control (SC); well-being (WB); sociability (SO); self-motivation (SM); adaptability (AD); self-acceptance (SA); positive relationships (PR); autonomy (AU); mastery of environment (ME); personal growth (PG); purpose in life (PL). * $p < 0.01$.

**Table 5.** Standardised regression weights for participants who have applied to 3 or more calls.

| Associations between Variables | RW | | | | SRW |
| --- | --- | --- | --- | --- | --- |
| | Estimations | SE | CR | *p* | Estimations |
| AN ← NES | 1.000 | | | | 0.833 |
| ST ← NES | 0.903 | 0.031 | 29.172 | ≤0.001 | 0.821 |
| DP ← NES | 0.938 | 0.035 | 27.027 | ≤0.001 | 0.741 |
| AD ← EI | 1.000 | | | | 0.201 |
| SM ← EI | 0.904 | 0.226 | 3.994 | ≤0.001 | 0.209 |
| SO ← EI | 1.795 | 0.373 | 4.808 | ≤0.001 | 0.452 |
| WB ← EI | 3.448 | 0.696 | 4.956 | ≤0.001 | 0.760 |
| SC ← EI | 1.862 | 0.390 | 4.772 | ≤0.001 | 0.404 |
| EM ← EI | 0.559 | 0.169 | 3.298 | ≤0.001 | 0.162 |
| SA ← PSWB | 1.000 | | | | 0.550 |
| PR ← PSWB | −0.289 | 0.060 | −4.849 | ≤0.001 | −0.140 |
| AU ← PSWB | 1.088 | 0.057 | 18.975 | ≤0.001 | 0.629 |
| ME ← PSWB | 0.597 | 0.046 | 12.886 | ≤0.001 | 0.402 |
| PG ← PSWB | 2.850 | 0.127 | 22.399 | ≤0.001 | 0.801 |
| PL ← PSWB | 1.551 | 0.069 | 22.486 | ≤0.001 | 0.840 |

**Table 5.** *Cont.*

| Associations between Variables | RW | | | | SRW |
| --- | --- | --- | --- | --- | --- |
| | Estimations | SE | CR | *p* | Estimations |
| EI ← NES | −0.011 | 0.004 | −3.055 | ≤0.001 | −0.029 |
| PSWB ← EI | 0.032 | 0.007 | 4.757 | ≤0.001 | 0.490 |
| PSWB ← NES | −0.086 | 0.008 | −11.439 | ≤0.001 | −0.440 |

**Note:** Depression (DP); anxiety (AN); stress (ST); emotionality (EM); self-control (SC); well-being (WB); sociability (SO); self-motivation (SM); adaptability (AD); self-acceptance (SA); positive relationships (PR); autonomy (AU); mastery of environment (ME); personal growth (PG); purpose in life (PL). **Note:** Regression weights (RW); standardised regression weights (SRW); standard error (SE); critical ratio (CR).

Comparing the effects between the two groups, a greater effect of negative emotions on emotional intelligence was noted for applicants who have undergone this process between zero and two times (β = −0.270; *p* ≤ 0.001 vs. β = −0.03; *p* ≤ 0.001). A smaller effect of negative emotional states on psychological well-being was observed for participants who have undergone this process more than twice (β = −0.570; *p* ≤ 0.001 vs. β = −0.440; *p* ≤ 0.001). Finally, a greater effect of emotional intelligence on psychological well-being was indicated for participants who have gone through the selection process between zero and two times (β = 0.653; *p* ≤ 0.001 vs. β = 0.490; *p* ≤ 0.001).

## 4. Discussion

The results show that there are differences between the two groups. This study shows that the teacher selection process influences the effects of the variables analysed.

In relation to the first research objective, the most recognised negative emotional state was stress. During the process of preparing for a test, stress leads to the appearance of burnout syndrome (Melguizo-Ibáñez et al. 2023; González-Valero et al. 2023). If these stress levels are not adequately treated, they can lead to increased levels of anxiety (Melguizo-Ibáñez et al. 2022). Depression was found to be the lowest-scoring emotional state. Depression levels tend to increase once the final evaluation of the process is known (Premkumar et al. 2022). If the degree of competence is low, depression levels will increase (Premkumar et al. 2022). Adequate emotional training has been found to help prevent burnout syndrome (Schoeps et al. 2019).

Regarding emotional intelligence, it was observed that the variable with the highest score was adaptability. It has been found that in preparation for such a demanding test, applicants carry out a process of adaptation (Carboneros-Castro et al. 2020; Mérida-López et al. 2023). During the adaptation process, some people show high levels of stress and anxiety due to the abrupt change in lifestyle (Amate-Romera and De la Fuente 2021). Educating students emotionally is one of the new educational demands (Gilar-Corbi et al. 2018). It has been shown that emotional training does not hinder the acquisition of competences and content (Gilar-Corbi et al. 2018). It does not interfere negatively with students' performance but, rather, helps to improve it (Gilar-Corbi et al. 2018).

For psychological well-being, the variables that showed the greatest recognition were personal growth and purpose in life. It has been noted that these two variables are positively related to the academic environment and, thus, to the process of preparing for the teacher selection process (Holzer et al. 2022). Obtaining a job as a public state teacher has a positive impact on eudemonistic well-being, since a desired goal is achieved (Holzer et al. 2022; Melguizo-Ibáñez et al. 2023).

Continuing with Objective 2 and Hypothesis 1, this hypothesis was not fulfilled. Differing results have been found by other studies (Tikkanen et al. 2022; Beck et al. 2013). The more times an assessment is taken, the greater the presence of anxiety and stress (Melguizo-Ibáñez et al. 2023). The teachers' examination process increases anxiety and stress levels (Melguizo-Ibáñez et al. 2022). The process of teachers' examinations can lead to the development of burnout syndrome (Zheng et al. 2022). This syndrome is characterised by the presence of high levels of

anxiety and stress (Carroll et al. 2021; Beames et al. 2023). This syndrome has a negative impact on psychological well-being and emotional intelligence (Lucas-Mangas et al. 2022). The results of this research show that participants with a lower number of attempts present a greater effect of anxiety and stress on emotional intelligence and psychological well-being. When carrying out this process, previous experience conditions emotional behaviours (Amate-Romera and De la Fuente 2021).

Continuing with Hypothesis 2, this hypothesis was not fulfilled. The start of preparation for an evaluative test originates with a positive attitude (Melguizo-Ibáñez et al. 2022). It has been observed that performance is conditioned by previous experience of taking a test (De la Fuente and Amate 2019). Experience has been shown to be a key element in dealing with evaluative tests (Kaminer et al. 2023). This experience translates into greater control of emotional competence (Squires et al. 2022; Pozo-Rico et al. 2023). The experience of performing this type of test has a positive effect on the control of emotions (De la Fuente and Amate 2019). The presence of negative emotions during the opposition process exerts a negative role on psychological well-being (Morales-Rodríguez et al. 2020).

This study is not free of limitations. The first of these is related to the type of study. The cross-sectional design only allowed us to establish the effects of the variables at the selected point in time. It would be interesting for future research to use a longitudinal design to measure the effect of the variables before and after the test. Similarly, the instruments showed a high degree of internal consistency, but they showed intrinsic error in the data measurement process. For future studies, it would be interesting to measure other academic variables and to include secondary school teaching applicants.

Although this study has limitations, the research shows strengths and a degree of applicability. The data were collected using instruments that have been validated and adapted by the scientific community. It should also be noted that the methods of data analysis are scientifically valid.

## 5. Conclusions

The present research investigated the levels of negative emotional states, emotional intelligence and psychological well-being, along with the effect of these variables as a function of the number of calls for applications to the teaching competitive examination process.

The descriptive analysis showed that anxiety and stress are the most common disruptive states. On the other hand, for emotional intelligence and psychological well-being, adaptability, personal growth and purpose in life are the factors that show the greatest recognition.

In the structural equation models, it was observed that a lower number of calls for applications leads to a greater effect of negative emotional states on emotional intelligence and psychological well-being. Therefore, it is concluded that the teacher selection process requires high emotional competence in order not to perceive a low degree of competence towards this form of access to the Spanish civil service.

The relevance of psychological well-being in the teacher selection process was thus shown. The relevance of anxiety, stress and depression in the teacher recruitment process was also highlighted. As a general conclusion, training in emotional competence during teacher training was noted. Through this training, teachers undergoing the selection process could be helped to reduce their negative emotional states. This would help them to achieve higher performance and improve their mental health.

**Author Contributions:** Conceptualisation, E.M.-I.; methodology, J.C.-Z. and J.L.U.-J.; software, E.M.-I. and G.G.-V.; validation, L.G.-P., P.P.-M. and J.L.U.-J.; formal analysis, E.M.-I.; investigation, J.C.-Z.; resources, G.G.-V.; data curation, J.L.U.-J.; writing—original draft preparation, L.G.-P.; writing—review and editing, P.P.-M.; visualisation, E.M.-I.; supervision, G.G.-V.; project administration, J.L.U.-J.; funding acquisition, J.L.U.-J. All authors have read and agreed to the published version of the manuscript.

**Funding:** This research received no external funding.

**Institutional Review Board Statement:** The study was conducted in accordance with the Declaration of Helsinki, and approved by the Ethics Committee of the University of Granada (2966/CEIH/2022, Approval Date: 27 September 2022).

**Informed Consent Statement:** Informed consent was obtained from all subjects involved in the study.

**Data Availability Statement:** The data used to support the findings of current study are available from the corresponding author upon request.

**Conflicts of Interest:** The authors declare no conflict of interest.

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
