# Peer review of "Emotional Status and Psychological Well-Being in the Educational Opposition Process"

_socsci, doi:10.3390/socsci12120685_

Round 1
Reviewer 1 Report
Comments and Suggestions for Authors
Dear Authors,
Thank you for the opportunity to read and review the manuscript "Emotional Status and Psychological Well-Being in the Educational Opposition Process ".
It is a very high-quality paper, which even in its current version is generally convincing in all areas. I congratulate the authors on this achievement!
I suggest that the theory, which is currently in the introduction, should be separated from this. This would create a separate theoretical chapter. The discussion can also be improved. It addresses fundamentally very important aspects, but is still superficial in some parts. Therefore, additions could be made incorporating the findings of other studies in order to emphasize the contribution of this study. The conclusions should also be deepened. What concrete implications can be derived for education policy? Here, too, it is important that you communicate and utilize the value of your study!
Author Response
Comment 1
Dear Authors,
Thank you for the opportunity to read and review the manuscript "Emotional Status and Psychological Well-Being in the Educational Opposition Process ".
It is a very high-quality paper, which even in its current version is generally convincing in all areas. I congratulate the authors on this achievement!
Response 1
Thank you very much for your comment.
The authors have tried our best to provide quality research.
Comment 2
I suggest that the theory, which is currently in the introduction, should be separated from this. This would create a separate theoretical chapter. The discussion can also be improved. It addresses fundamentally very important aspects, but is still superficial in some parts. Therefore, additions could be made incorporating the findings of other studies in order to emphasize the contribution of this study. The conclusions should also be deepened. What concrete implications can be derived for education policy? Here, too, it is important that you communicate and utilize the value of your study!
Response 2
Thank you very much for your comment. All requested changes have been implemented. The authors believe that your suggestions have helped to improve the quality of the research.
Reviewer 2 Report
Comments and Suggestions for Authors
The study is relevant in content but insufficient in its execution. Here are some notes:
- The same references are repeated several times within a short distance of each other (e.g., De la Fuente and Amate, 2019; Goleman, 1995; Tomas et al., 2016). I recommend considering mentioning them only once, modifying the sentence construction accordingly.
- It seems to me that the literature cited in the Introduction is very limited, compared to the countless studies that have been carried out over the years on teachers and their well-being or work-related stress. I find that the analysis of the literature, in general, has been very summarized, considering the vastness and complexity of the variables examined. I would recommend that authors carry out a broader and more focused analysis of the literature (the same papers are cited over and over again).
- The description of the sample needs to be added: how much percentage of males and females? Also, I would find it more interesting to know the distribution of the sample by the number of examinations undertaken, rather than by regions of Spain (not very useful for the research objectives and of little interest to non-Spanish readers).
- I recommend adding some example items for each scale, so that the reader can get an idea of the operationalization of each construct.
- The methodology chosen to test the model (i.e., the dimensions that make up the variable saturating on the factor and comparing the factors as unique measures) means that a great deal of information is lost. For example, the relationships between, respectively, anxiety, stress and depression and each dimension of psychological well-being. Which of the three disruptive states weighs the most? Impossible to say at present. I would ask the authors at least for an explanation as to why they chose to test the model in this way. Clearly, redoing the analyses would be ideal and would greatly benefit their study.
- Furthermore, since it is a mediation model, the authors forgot to report the indirect effects.
- Something is not right in the discussion of the results in lines 33-40. The authors state: “Comparing the effect of the variables as a function of the number of calls, a greater effect is observed for participants who have taken the test more than three times for negative emotional states on emotional intelligence (β=−0.03 p < 0.01; β=−0.27 p < 0.01). A greater effect is observed for participants who have taken the test more than 3 times of negative emotional states on psychological well-being (β=−0.44 p < 0.01 vs β=−0.57; p < 37 0.01).” However, it is exactly the opposite, given that the most negative effects belong to figure 2, i.e. the one relating to participants who showed up 0 to 2 times. Therefore, the hypotheses are not confirmed. According to the values reported in Figure 2, candidates who sat fewer than twice in the teacher competition process showed a stronger effect of disruptive states on psychological well-being and emotional intelligence.
- Based on all the methodological issues highlighted so far, the entire discussion falls apart. The conclusions are not supported by the data or the way the model was tested.
Extensive English revision is necessary.
Author Response
Comment 1
The study is relevant in content but insufficient in its execution. Here are some notes:
- The same references are repeated several times within a short distance of each other (e.g., De la Fuente and Amate, 2019; Goleman, 1995; Tomas et al., 2016). I recommend considering mentioning them only once, modifying the sentence construction accordingly.
- It seems to me that the literature cited in the Introduction is very limited, compared to the countless studies that have been carried out over the years on teachers and their well-being or work-related stress. I find that the analysis of the literature, in general, has been very summarized, considering the vastness and complexity of the variables examined. I would recommend that authors carry out a broader and more focused analysis of the literature (the same papers are cited over and over again).
Response 1
Thank you very much for your comments. The authors have carried out a further literature review. We have added new citations that reinforce the ideas.
Comment 2
The description of the sample needs to be added: how much percentage of males and females? Also, I would find it more interesting to know the distribution of the sample by the number of examinations undertaken, rather than by regions of Spain (not very useful for the research objectives and of little interest to non-Spanish readers).
I recommend adding some example items for each scale, so that the reader can get an idea of the operationalization of each construct.
Response 2
Thank you very much for your suggestion. The authors have considered keeping table 1. We have also added the distribution according to gender and according to the calls submitted. We have also added some examples of the items in each questionnaire.
Comment 3
The methodology chosen to test the model (i.e., the dimensions that make up the variable saturating on the factor and comparing the factors as unique measures) means that a great deal of information is lost. For example, the relationships between, respectively, anxiety, stress and depression and each dimension of psychological well-being. Which of the three disruptive states weighs the most? Impossible to say at present. I would ask the authors at least for an explanation as to why they chose to test the model in this way. Clearly, redoing the analyses would be ideal and would greatly benefit their study.
Furthermore, since it is a mediation model, the authors forgot to report the indirect effects.
Response 3
Thank you very much for your comment. The authors fully understand your point of view.
The authors have carried out new theoretical modelling. Initially we have treated the variables in a unidimensional perspective. This has yielded lower values than those proposed by Maydeu-Olivares, 2017 and Kyriazos, 2018 (X2/gl=102.398, RMSEA=1.192, CFI=0.198; AGFI=0.364; GFI=0.334; TLI=0.248; NFI=0.587). These indices do not guarantee the quality of the proposed model. Subsequently, we have treated each of the variables that make up emotional intelligence, well-being and negative emotional states individually. The following fit values were obtained: (X2/gl=132.589, RMSEA=1.634, CFI=0.245; AGFI=0.318; GFI=0.649; TLI=0.741; NFI=0.652). With the proposed analysis, the adjustment values are in line with the proposed benchmarks. Similarly, the proposed models show the largest effect of each grouping variable (Negative Emotional States, Psychological Well-Being and Emotional Intelligence). The new tables (4 and 5) allow us to study the sub-variable that exerts the greatest effect on each sub-variable.
Having said that, the explanation you ask for to justify the model is mainly due to the values obtained for the different fit indices. A well-fitted model gives satisfactory answers to the research objectives and hypotheses. Once the theoretical model has been developed with good fit indices, the data obtained are fully reliable.
Following your last comment, (the authors forgot to report the indirect effects) the tables provided by the statistical programme used have been added (Table 4 and Table 5).
Comment 4
Something is not right in the discussion of the results in lines 33-40. The authors state: “Comparing the effect of the variables as a function of the number of calls, a greater effect is observed for participants who have taken the test more than three times for negative emotional states on emotional intelligence (β=−0.03 p < 0.01; β=−0.27 p < 0.01). A greater effect is observed for participants who have taken the test more than 3 times of negative emotional states on psychological well-being (β=−0.44 p < 0.01 vs β=−0.57; p < 37 0.01).” However, it is exactly the opposite, given that the most negative effects belong to figure 2, i.e. the one relating to participants who showed up 0 to 2 times. Therefore, the hypotheses are not confirmed. According to the values reported in Figure 2, candidates who sat fewer than twice in the teacher competition process showed a stronger effect of disruptive states on psychological well-being and emotional intelligence.
Response 4
Thank you very much for your review. The authors agree with you. The errors in the areas noted have been corrected.
Comment 5
Based on all the methodological issues highlighted so far, the entire discussion falls apart. The conclusions are not supported by the data or the way the model was tested
Response 5
Thank you very much for your comment.
The authors disagree with this comment. At all times, the criteria used to develop the analysis carried out have been the criteria collected in different research specialised in this type of analysis. Likewise, the models developed show a good fit in each of the different indices.
Round 2
Reviewer 2 Report
Comments and Suggestions for Authors
The authors have modified and improved the previous version as much as possible, however there is still work to be done in the Discussion paragraph.
The discussion from line 72 to line 92 is inconsistent and fragmentary, which depends partly on the construction of the sentences and partly on the concepts expressed. The authors correctly modified statements about hypotheses that were not confirmed. However, I expected them to give explanations and hypotheses as to why they got these results and not the ones they expected. What could be the reason why teachers who have participated in the process the fewest times are the most stressed? And so on. It is necessary to explain in more depth the implications of the results obtained and launch hypotheses, in light of the existing literature and the authors' thoughts.
Finally, another small error remained in line 93, which still states "despite having met the research objectives and hypotheses" even if the hypotheses were not confirmed.
Comments on the Quality of English LanguageAn English revision is needed.
Author Response
Comment 1
The authors have modified and improved the previous version as much as possible, however there is still work to be done in the Discussion paragraph.
The discussion from line 72 to line 92 is inconsistent and fragmentary, which depends partly on the construction of the sentences and partly on the concepts expressed. The authors correctly modified statements about hypotheses that were not confirmed. However, I expected them to give explanations and hypotheses as to why they got these results and not the ones they expected. What could be the reason why teachers who have participated in the process the fewest times are the most stressed? And so on. It is necessary to explain in more depth the implications of the results obtained and launch hypotheses, in light of the existing literature and the authors' thoughts.
Response 1
Thank you very much for your comments. The authors agree with you. We have proceeded to reformulate the lines you have pointed out.
Comment 2
Finally, another small error remained in line 93, which still states "despite having met the research objectives and hypotheses" even if the hypotheses were not confirmed.
Response 2
Thank you very much for your comment.
The authors agree with you completely. We have removed this statement.
Regarding the writing of the research, it has been reviewed by an English expert